# Studies on Simultaneous Enrichment and Detection of *Escherichia coli* O157:H7 during Sample Shipment

**DOI:** 10.3390/foods11223653

**Published:** 2022-11-15

**Authors:** Chuyan Chen, Claudia P. Coronel-Aguilera, Bruce M. Applegate, Andrew G. Gehring, Arun K. Bhunia, George C. Paoli

**Affiliations:** 1Department of Food Science, Purdue University, West Lafayette, IN 47907, USA; 2Department of Biological Sciences, Purdue University, West Lafayette, IN 47907, USA; 3Center for Food Safety Engineering, Purdue University, West Lafayette, IN 47907, USA; 4Purdue Institute of Inflammation, Immunology and Infectious Disease, Purdue University, West Lafayette, IN 47907, USA; 5Interdisciplinary Life Science Program (PULSe), Purdue University, West Lafayette, IN 47907, USA; 6Department of Comparative Pathobiology, Purdue University, West Lafayette, IN 47907, USA; 7Molecular Characterization of Foodborne Pathogens Research Unit, Eastern Regional Research Center, Agricultural Research Service, U.S. Department of Agriculture, Wyndmoor, PA 19038, USA

**Keywords:** *E. coli* O157:H7, Shiga toxin-producing *E. coli*, bacteriophage, culture enrichment, foodborne pathogen, pathogen detection

## Abstract

The USDA-FSIS has zero tolerance for *E. coli* O157:H7 in raw ground beef. Currently, FSIS collects samples from beef processing facilities and ships them overnight to regional testing laboratories. Pathogen detection requires robust methods that employ an initial 15–24 h culture enrichment. This study assessed the potential of using the ΦV10*nluc* phage-based luminescence detection assay during enrichment while the sample is in transit. Parameters including phage concentrations, temperature, and media-to-sample ratios were evaluated. Results in liquid media showed that 1.73× 10^3^ pfu/mL of ΦV10*nluc* was able to detect 2 CFU in 10 h. The detection of *E. coli* O157:H7 was further evaluated in kinetic studies using ratios of 1:3, 1:2, and 1:1 ground beef sample to enrichment media, yielding positive results for as little as 2–3 CFU in 325 g ground beef in about 15 h at 37 °C. These results suggest that this approach is feasible, allowing the detection of a presumptive positive upon arrival of the sample to the testing lab. As the current cargo hold controlled temperature is required to be 15–25 °C, the need for elevated temperature should be easily addressed. If successful, this approach could be expanded to other pathogens and foods.

## 1. Introduction

Foodborne pathogens cause an estimated 9.4 million cases of illness annually in the United States. Of those, there are over 175,000 cases of illnesses caused by Shiga toxin-producing *Escherichia coli* (STEC) infections [1]. STEC commonly cause gastroenteritis and those that result in enterohaemorrhagic gastroenteritis are called (EHEC). The EHEC pathovar has a known pathology by attaching and effacing (A/E) lesions on intestinal epithelial cells, destroying microvilli and inhibiting actin function, and forming pedestals below the site of attachment. Infection by EHEC manifests into symptoms such as bloody diarrhea [2]. STEC infections are defined by the specific production of one or both Shiga toxins (Stx_1_ or Stx_2_), causing symptoms that range from mild to life-threatening, including stomach cramps, vomiting, and diarrhea [3]. *E. coli* O157:H7 infections are a main cause of hemolytic uremic syndrome, which is associated with severe health implications, including kidney failure, hemolytic anemia, and potential death [4]. The primary reservoir for STEC has been recognized to be cattle [5]. From 1982 to 2002, outbreaks found to be associated with the Shiga toxin producing *E. coli* O157:H7 were primarily foodborne, transmitted by ground beef [6]. In addition, other sources of outbreaks have involved raw milk, vegetables, sandwiches, and water [7,8,9,10].

Since 1994, STEC O157:H7 has been under strict “zero-tolerance” enforcement in ground beef by the U.S. Department of Agriculture (USDA) Food Safety and Inspection Service (FSIS) [11]. This “zero-tolerance” policy means that any sample testing positive for STEC O157;H7 (even a single cell in a 325 g sample) is considered adulterated and not fit for consumption. In 2012, this policy expanded to include the six serotypes of non-O157 STEC, due to their increasing prevalence in causing foodborne illness in the U.S. [12]. With a low infectious dose of an estimated 10–100 cells, *E. coli* O157:H7 poses a considerable threat to public health [13,14]. Thus, it is critical that *E. coli* O157:H7 detection in food is carried out rapidly and accurately.

Traditional methods for detection of STEC O157:H7 and other foodborne pathogens involve culture media enrichment, isolation on solid selective/differential media, and biochemical identification of isolates, which can take several days to weeks to complete. Over the past few decades, numerous rapid methods have been developed that are typically immunologically (antibody)-based or nucleic acid-based (e.g., PCR) [15,16].

While these methods have grown very popular, government regulatory agencies, such as the USDA-FSIS, continue to employ microbiological methods to acquire the required bacterial isolate. For example, the FSIS method for detection of STEC O157:H7 and other regulated STEC in meat products involves an initial 15–24 h culture enrichment of a 325 g sample in 975 mL of modified tryptone soy broth (mTSB) [17,18], screening of the enrichment by PCR, separation of STEC by immunomagnetic separation, plating on modified Rainbow Agar, and testing isolated by O-antigen agglutination to acquire a presumptive positive isolate. Presumptive positive isolates are then grown on sheep blood agar, tested again for O antigen agglutination, identified using the Bruker MALDI Biotyper, and once again screened by PCR [17].

In October 1994, the USDA-FSIS commenced a verification sampling program for *E. coli* O157:H7 in order to verify process control, to stimulate industry testing, and to reduce the pathogen presence in raw ground beef. Over time, the testing program expanded to include other beef parts, such as trim and other components. In 2010, over 85% of samples analyzed were of raw ground beef [19], and in 2022, more than half of beef samples tested (62%) still consisted of ground beef [20]. Food samples for testing at FSIS labs are collected at the production facility by FSIS inspectors. The inspectors randomly and aseptically collect raw ground beef samples from the current day’s production. The raw ground beef samples (325 g each) are placed into three Whirl-Pak bags, chilled, and sent to the testing laboratory. Assuming samples are shipped at 5 PM daily and received 8 AM the following morning, 15 h is the minimum time that a ground beef sample will spend in transit. The testing laboratory uses one or two samples for pathogen testing, and the third is held for additional analysis in the event of a positive sample [21]. 

We have previously described a bacteriophage with an integrated luminescence reporter for the detection of *E. coli* O157:H7 [22]. This reporter phage ΦV10*nluc* was constructed by cloning the genes for NanoLuc^®^ luciferase [23] into the *E. coli* O157:H7-specific temperate phage ΦV10 [24,25]. NanoLuc^®^ produces ATP-independent luminescence after reacting with furimazine substrate. While bacterial luciferase is 77 kDa, NanoLuc^®^ luciferase is a small protein subunit which is only 19 kDa [26]. Its smaller size gives it flexibility in bioreporter construction. The light intensity produced by NanoLuc^®^ has been found to be roughly 150-fold greater than either Firefly or *Renilla* luciferases [23]. The ΦV10*nluc* reporter is added during culture growth, and is capable of detecting as few as five cells of *E. coli* O157:H7 in 40 mL of ground meat slurry within 9 h [22]. 

This study assessed the potential of using the ΦV10*nluc* reporter system to detect *E. coli* O157:H7 in mTSB, leveraging the 15 or more hours that the sample spends in transit from the production facility to the FSIS lab. The potential to exploit shipment time for enrichment could allow presumptive positive samples to be rapidly identified soon after samples are received in the testing laboratories. In order to assess the potential of this method, key factors including phage concentrations, temperature, and sample to media ratios were evaluated. 

In addition, prior research indicated that ΦV10*nluc* phage preparations may be contaminated with NanoLuc^®^ luciferase released from the cells during lytic propagation. This co-purification of phage and luciferase results in a significant background luminescence upon addition of the enzymatic substrate. In this study, a previously reported method for phage purification was modified and evaluated for its efficacy in removing the luciferase contaminant.

## 2. Materials and Methods

### 2.1. Media Preparation

All media components were purchased from Fisher Scientific (Waltham, MA, USA) unless otherwise noted. Lysogeny broth (LB) was prepared by adding 10 g of tryptone powder, 10 g of sodium chloride, and 5 g of yeast per liter of deionized water. All LB media were adjusted to a pH of 7.5. The LB plates and top agar were made by adding 17 g and 6 g, respectively, of agar to 1 L of the LB media. When used, kanamycin (IBI Scientific, Las Vegas, NV, USA) was added to the LB plates at a final concentration of 50 µg/mL. Modified tryptone soya broth (mTSB) was made according to the USDA-FSIS Media and Reagents guideline (USDA-FSIS 2022). 

### 2.2. E. coli O157:H7 Culture Preparation

The *E. coli* strain C7927 used in this study is a human STEC serotype O157:H7 isolate from an apple cider outbreak [27]. A single colony of *E. coli* O157:H7 C7927 from an LB agar plate was inoculated in 100 mL of LB liquid media and grown overnight at 37 °C with shaking (100 rpm). After overnight incubation, 100 µL of serial dilutions (1:10) were spread onto LB agar plates to enumerate the number of colony-forming units (CFU)/mL of the overnight culture.

### 2.3. Phage Purification Procedure

In order to prepare the bacteriophage, 50 μL of a ΦV10*nluc* lysogen of *E. coli* O157:H7 C7927 [22] was inoculated in one liter of LB containing kanamycin (50 µg/mL) and incubated overnight on a shaker at 100 rpm at 37 °C. Following overnight incubation, 4 mL of chloroform (Fisher Scientific, Waltham, MA, USA) was added to the culture in order to permeabilize the cell membrane and to complete cell lysis. In addition, 48 g of sodium chloride was added and dissolved with constant stirring. The solution was then centrifuged at 15,000× *g* for 10 min in order to collect bacteria and cell debris into a pellet. The supernatant containing the phage was collected, and 80 g of polypropylene glycol (Fisher Scientific, Waltham, MA USA) per liter was dissolved into the solution using slow agitation. The solution was stored at 4 °C overnight to precipitate the phage. The phage solution was then centrifuged at 17,000× *g* for 15 min in order to collect the phage. The supernatant was carefully decanted, and the phage pellet was resuspended in phage buffer (50 mM Tris, 100 mM MgCl_2_, pH 7.6) and vacuum filtered through a 0.45 μm pore size membrane (Fisher Scientific, Waltham, MA, USA). Subsequently, the filtrate was passed through a second filter with pore size of 0.1 μm (Fisher Scientific, Waltham, MA, USA). The resulting solution was collected and stored in a sterile 50 mL Falcon screw cap tube at 4C. 

When using the ΦV10*nluc* phage prepared as described above, phage infection assays yielded significant background light upon addition of the Nano-Glo^®^ reagent (Promega Madison WI; see Section 2.5.). Furthermore, addition of the Nano-Glo^®^ reagent directly to the ΦV10*nluc* phage preparations resulted in light emission, suggesting the presence of free luciferase protein in the phage preparations. Thus, a subsequent purification procedure was conducted to remove the free luciferase and reduce the background light emission from the ΦV10*nluc* phage preparations, as briefly described here. An Amicon^®^ Ultra-15 (MilliporeSigma, Burlington, MA, USA) centrifugal filter device that contained a membrane with a 100,000 kDa cutoff was used to separate the luciferase from the ΦV10*nluc* phage. Twelve (12) mL of phage preparation was added to each centrifugal filter device, which was then placed in a fixed-angle rotor centrifuge and spun at 5000× *g* for 15 min per cycle, for a total of 7 cycles. Since the NanoLuc^®^ (19 kDa) protein is smaller in size compared to the phage (>100 kDa), the phage is concentrated on the filter while the protein is washed out through the filter. After every cycle the filtrate was measured for NanoLuc^®^ luminescent activity (described below) and the resuspended phage concentration was determined. The purified and concentrated ΦV10*nluc* phage was then resuspended in phage buffer for storage at 4 °C until further analysis. 

2.4. ΦV10nluc Phage Titer Determination

Coomassie Brilliant Blue G-250 dye (Bio-Rad laboratories, CA USA) was added at 1% wt./vol to top agar in order to enhance the contrast of phage plaques, increasing their visibility and making them easier to count [22]. For plaque assays, 200 μL of an overnight culture of *E. coli* O157:H7 C7927 and 100 μL of serial dilutions of the ΦV10*nluc* phage solution were added to melted blue top agar, vortexed, and then poured onto LB plates. Plates were incubated in a 37 °C incubator for 18 h and plaques were enumerated the next day for phage titer determination. Plaque assays were conducted in triplicate. 

### 2.5. Nano-Glo^®^ Reagent Preparation

The Nano-Glo^®^ reagent (Promega, Madison, WI, USA), which produces ATP-independent luminescence upon the oxidation of furimazine by the NanLuc luciferase, was used to measure presence of NanLuc luciferase according to the instructions from the manufacturer. An aliquot of 20 μL of Nano-Glo^®^ substrate was added to 1 mL of Nano-Glo^®^ buffer. The resulting reagent was vortexed and either used immediately or stored at 4 °C for later use.

### 2.6. Effects of Phage Concentration on Time to Detection 

Initial experiments were performed in LB to determine the relationship between phage concentrations (10^2^–10^5^ pfu/mL) and time to detection with *E. coli* O157:H7, ranging from approximately 2 to 2 × 10^5^ CFU per assay. Each assay consisted of a total of 40 mL.

Ten-fold dilutions of cells from 10^S5^ to 10^−11^ were prepared from an overnight culture for phage assays. One hundred microliters of each dilution were spread on LB plates in triplicate and incubated overnight at 37 °C to determine the initial number of CFU/mL in the overnight culture.

### 2.7. Characterization of the Growth of *E. coli* O157:H7 and Corresponding ΦV10nluc Lysogen

Growth of the wild-type *E. coli* O157:H7 C7927 and the C7927 ΦV10*nluc* lysogen was characterized by measuring the OD_600_ of cultures grown in mTSB. A colony of C7927 ΦV10*nluc* lysogen was tested in 1 mL of LB and 10 μL Nano-Glo^®^ reagent for luminescence prior to inoculation of an overnight growth. A 20 μL aliquot of an overnight culture of wild-type and ΦV10*nluc* lysogen strains was inoculated in 100 mL mTSB flasks, then incubated at 37 °C with shaking at 100 rpm. Growth curves were performed in triplicate. Optical density measurements (OD_600_) were taken using a BioPhotometer (Eppendorf North America, Enfield, CT, USA) at inoculation and every 25 min for 8 h. 

### 2.8. E. coli O157:H7 Detection in Raw Ground Beef

In order to evaluate the ability of the purified ΦV10*nluc* phage to detect *E. coli* O157:H7 cells in raw ground beef, matrices assays were performed following standard and modified FSIS protocols. Approximately 1.73 × 10^3^ pfu/mL final phage concentrations were used with ground beef in stomacher bags and Nalgene bottles, with a final volume of 1 L. This concentration was achieved by the addition of 100 μL of 1.73 × 10^7^ pfu/mL phage per assay. All raw ground beef used in this study was purchased at a local grocery store and consisted of 83% lean meat, 17% fat. All ground beef was used immediately after purchase. 

#### 2.8.1. Detection of E. coli O157:H7 in Raw Ground Beef in Stomacher Bags

Samples of 325 g of ground beef were put in 975 mL of mTSB broth (1:3 sample to enrichment media) inside a sterile, plain, clear polypropylene stomacher bag without a filter mesh. *E. coli* O157:H7, ranging from approximately 3 to 3 × 10^4^ CFU per bag, were inoculated into each ground beef slurry. Ground beef with mTSB broth and wild-type ΦV10 phage was used as the negative control. Each sample was hand-massaged for 30 s at the initial time of inoculation. Each assay was performed in triplicate. Luminescence measurements of 1 mL samples were taken after the addition of 10 μL of previously prepared Nano-Glo^®^ reagent using a Sirius luminometer (Berthold Detection Systems, Bad Wildbad, Germany) once per hour. 

#### 2.8.2. Detection of E. coli O157:H7 in Raw Ground Beef in Nalgene Bottles

In order to determine the efficacy of the phage-based detection during shipping of ground beef samples, different ratios of beef sample to mTSB were evaluated in selected sample containers. I-chem Brand N311-1000 Nalgene 300 Series HDPE Wide Mouth Bottles (Thermo Fisher Waltham, MA, USA) were chosen due to size, low leak potential, and pressure requirements for shipping.

Initial experimentation included 325 g of raw ground beef added to 650 mL of mTSB for a 1:2 sample to media ratio. The same amount of raw ground beef was used with 325 mL of mTSB for a separate experiment at a 1:1 ratio. Each sample was inverted by hand for 30 s at the initial time of inoculation. All Nalgene bottles used were sterile. *E. coli* O157:H7 cell inocula were approximately 3 to 3 × 10^2^ CFU per bottle in the 1:2 sample to media ratio and approximately 2 to 2 × 10^2^ CFU per bottle in the 1:1 sample to media ratio.

## 3. Results and Discussion

### 3.1. Phage Purification

When the bacteria are infected by ΦV10*nluc*, the cells may be lysed or become lysogens. When the lysogens are stressed by environmental factors, they may go into the lytic phase. They can spontaneously go lytic as well, or, during preparation, the addition of chloroform and sodium chloride can affect the lytic process. While intact and injured cells and cell debris are removed by centrifugation and filtration in the initial phage preparation, the phage as well as the NanoLuc^®^ luciferase remain in solution, resulting in high levels of background light emission upon addition of the Nano-Glo^®^ substrate. The presences of the NanoLuc^®^ luciferase resulted in high background luminescence in *E. coli* O157:H7 detection assays. In an attempt to remove the NanoLuc^®^ luciferase from the phage, three purified phage preparations were subjected to further purification using Amicon^®^ Ultra-15 centrifugal filtration devices [28]. Removal of the NanoLuc^®^ luciferase was determined by measuring the luminescence generated by the Nano-Glo^®^ reagent in the filtrate after each centrifugation/filtration cycle. Reduced light emission in sequential rounds of filtration demonstrated the removal of the luciferase from the phage preparation (Table 1). The luciferase activity in the filtrate was reduced by approximately five orders of magnitude through the first six cycles of filtration, but plateaued to about two to four thousand RLUs after the seventh round of filtration (Table 1). The total background light emission in the phage retentate was also significantly reduced by roughly five orders of magnitude, from an average of 7.03 × 10^8^ RLU/s to 9.33 × 10^3^ RLU/s per 10 μL sample assay (Table 2). The residual light generated after filtration appeared to indicate physical association of luciferase with the phage. Although this background light emission is not negligible, it is sufficiently low as to not interfere with *E. coli* O157:H7 detection assays and may serve as an indicator that the ΦV10*nluc* phage was indeed added to negative samples. It is worth noting that there was a consistent 10- to 100-fold reduction in phage concentration after the filtration steps. Plaque assays confirmed that a minimal amount of phage was found in the filtrate at the end of the purification procedure, suggesting that the phage was not being lost through the membrane.

### 3.2. E. coli O157:H7 Detection by ΦV10nluc Phage in LB

The ability of various concentrations of ΦV10*nluc* phage to detect *E. coli* O157:H7 cells was assessed in LB. These experiments were performed in order to test whether a higher concentration of ΦV10*nluc* phage would contribute to a shorter time to detection than previously shown by Zhang et al. [22]. ΦV10*nluc* phage concentrations ranging from 10^2^ to 10^5^ pfu/mL were assessed for time to detection of 0 to 10^5^ *E. coli* O157:H7 CFU per assay. Results shown in Figure 1A indicate that 1.76 × 10^2^ pfu/mL phage concentration was able to detect the presence of approximately 2 CFU in 40 mL of LB in the shortest amount of time—just over 6 h with no prior incubation. Additionally, the 10^3^ pfu/mL phage concentration detected approximately 2 cells in under 10 h, while 10^4^ phage concentration was able to detect the same in roughly 8.5 h. ΦV10*nluc* phage concentration of 4.5 × 10^5^ pfu/mL was able to detect approximately 200 cells in roughly 9 h (Figure 1C). The time to detection using both 4.33 × 10^3^ pfu/mL and 4.33 × 10^4^ pfu/mL phage concentrations decreased as cell numbers increased (Figure 1A–C). The phage concentration of 4.33 × 10^4^ pfu/mL could detect approximately 2 cells in 8.5 h, 20 cells at 7 h, and 200 cells at 6 h. The phage of 4.33 × 10^3^ pfu/mL detected approximately 2 cells in just under 10 h, while it was able to detect approximately 20 cells in 8.5 h, and approximately 200 cells in just under 7 h. ΦV10*nluc* of 1.73 × 10^3^ pfu/mL concentration was used in subsequent experiments due to short time to detection and a low background luminescence. In addition, a higher phage concentration did not necessarily correlate with lower time to detection, exemplified by the inability of 10^5^ pfu phage to detect either 2 or 20 cells within 12 h (Figure 1A,B).

### 3.3. Characterization of the Growth of E. coli O157:H7 and ΦV10nluc Lysogen

The growth of *E. coli* O157:H7 was compared with the ΦV10*nluc* lysogen growth in mTSB at 37 °C by measuring culture optical density. The wild-type *E. coli* O157:H7 culture has an approximate growth of 0.87 OD_600_/h, while the ΦV10*nluc* lysogen has an estimated growth rate that is less than half of that of the wild-type, at 0.38 OD_600_/h (Figure 2). After the initial lag of about 3.5 h, the optical density of *E. coli* O157:H7 cells doubled roughly every 50 min, while the ΦV10*nluc* lysogens showed a lag of about 6 h and an estimated doubling time of approximately 110 min. At room temperature (25 °C), the growth rate of the wild-type *E. coli* O157:H ΦV10*nluc* lysogen decreased to about 0.26 OD_600_/h (doubling time of ~160 min) after a lag of approximately 20 h (Figure 3). 

### 3.4. E. coli O157:H7 Detection in Raw Ground Beef

The concentration of 1.73 × 10^3^ pfu/mL of ΦV10*nluc* was used to evaluate time to detection of varying amounts of *E. coli* O157H7 C7927 in raw ground beef slurry. First, the FSIS protocol was followed using 325 g inoculated raw ground beef containing added phage in 975 mL of mTSB enrichment media within stomacher bags. Luminescence measurements were recorded hourly for 15 h and are shown in Figure 4. Increased light production correlated to increasing cell concentrations.

### 3.5. Growth of E. coli O157:H7 C7927

It is worth noting that the luminescence in the control samples not inoculated with *E. coli* O157:H7 gradually decreased over time. This gradual decrease was not observed in control experiments conducted in LB, suggesting this phenomenon may be due to proteases naturally present in the raw ground beef that may be degrading the NanoLuc^®^ luciferase, leading to lower observed background luminescence. In addition, the source of variability in time to detection is most likely due to the increased enrichment volume (1300 mL) compared to the previous report by Zhang et al. [22], in which 40 mL ground beef slurries were used. For comparison, the cell concentration in the previously reported 40 mL assays [22] would be approximately equivalent to the 30 CFU inoculation shown in Figure 4.

### 3.6. Detection of E. coli O157:H7 in Raw Ground Beef in Nalgene Bottles

Due to the potential for leaks using an enrichment volume of 1300 mL, enrichment during shipment is not reasonably conducted in stomacher bags; therefore, the capability of the ΦV10*nluc* phage to detect *E. coli* O157:H7 in one-liter Nalgene bottles was investigated. One-liter bottles were chosen due to cost and shipment weight. Previous work by Cai and Cabezas ([29], unpublished) demonstrated positive results with ground meat samples which were simply sprayed with ΦV10*nluc*. This suggested that we could combine the use of the one-liter Nalgene bottles with lowered mTSB media volume (650 mL, 325 mL) with the ground meat sample (325 g), which would facilitate enrichment during transport (Figure 5).

Ground beef slurries (650 mL mTSB and 325 g ground beef) in Nalgene bottles containing approximately 0, 3, 30, and 300 CFU of *E. coli* O157:H7 were subjected to the phage-based luminescence assay. In all three experimental replicates, 300 CFU were detected at roughly 11 h (Figure 6). Results also showed that 30 CFU was detected in all three replicates in roughly 14 h. However, for three CFU, only the first replicate resulted in detection at about 15 h (Figure 6A). This fractional positive result at low inoculation levels is not surprising, given the likelihood that some samples may not have been inoculated with any cells. 

In this study, pressure was observed to have built up in the bottles during enrichment. This may be due to the shortage of dissolved and headspace oxygen initially available within the bottle, resulting in fermentative growth and gas production by *E. coli* O157:H7 or the beef microbiota. Though no leaks were observed in this study, this observation raises a potential safety concern related to sample integrity (i.e., bottle leakage) for enrichment during shipment.

An additional experiment was conducted using a 1:1 ratio of mTSB media to raw ground beef in the Nalgene bottles over 20 h in order to evaluate if a decrease in media would lead to higher available oxygen for the bacteria, potentially resulting in a shorter time to detection. Observations during this experiment included that there was still gas release upon opening the bottles after 8 h; however, there was less pressure and less gas released compared with the 2:1 ratio experiment. Results indicate that the ΦV10*nluc* phage detected approximately two cells in one out of three assays (Figure 7B). Figure 7 shows that a higher cell concentration correlated to higher measured luminescence over time. Similar to the 2:1 ratio Nalgene bottle experiment, 200 CFU was detected at roughly 11 h in all 3 replications while 20 CFU was detected at roughly 15 h. No luminescence above the control response was detected in any Nalgene bottle studies containing 0.2 to 0.3 CFU (Figure 6 and Figure 7). 

## 4. Conclusions

Previous work has confirmed ΦV10*nluc* specificity to *E. coli* O157:H7 isolates [22]. This research showed that ΦV10*nluc* phage can detect roughly five *E. coli* O157:H7 cells in 40 mL raw ground beef slurry after approximately 9 h of enrichment/incubation. Such performance indicates the potential for exploiting this phage infection system as a detection platform coincident with ground beef regulatory test sample shipment. However, the previous work conducted in 40 mL assays was not representative of current FSIS sampling methods. Therefore, this study evaluated the FSIS-recommended sample to media volume of a 325 g raw ground beef sample in 975 mL of mTSB enrichment media, and showed that ΦV10*nluc* was able to detect *E. coli* O157:H7, showing higher cell concentrations correlated with high luminescence in real-life application sample sizes. Within government regulatory protocols which are recommended to ensure food safety, the enrichment step is ubiquitous across all official and rapid methods used to detect foodborne pathogens. While culture enrichment can take up to 24–48 h after receipt of the sample at the regulatory lab, there is an opportunity to exploit this time during shipping for sample enrichment. Shipment of samples from meat production facilities to testing laboratories takes approximately 15–18 h. While it is known that FedEx Express Controlled Room Temperature holds samples at 15–25 °C during shipping, growth curves in this study show that this phage-based assay will not function optimally even at the upper limit of the cargo hold of 25 °C. Growth at 37 °C will be necessary to maximize the growth of STEC and, at least partially, to limit the growth of some of the foodborne microbiota. Competition studies are recommended to evaluate how background microbiota may affect the sensitivity of this detection method. In addition, the temperature recommendation of 37 °C for optimal performance should be addressed through further engineering studies. Potentially, a simple and inexpensive heat source, akin to a dry chemical or gel pack handwarmer or insulated boxes, may be assessed for maintaining the required temperature of samples during shipment.

## Figures and Tables

**Figure 1 foods-11-03653-f001:**
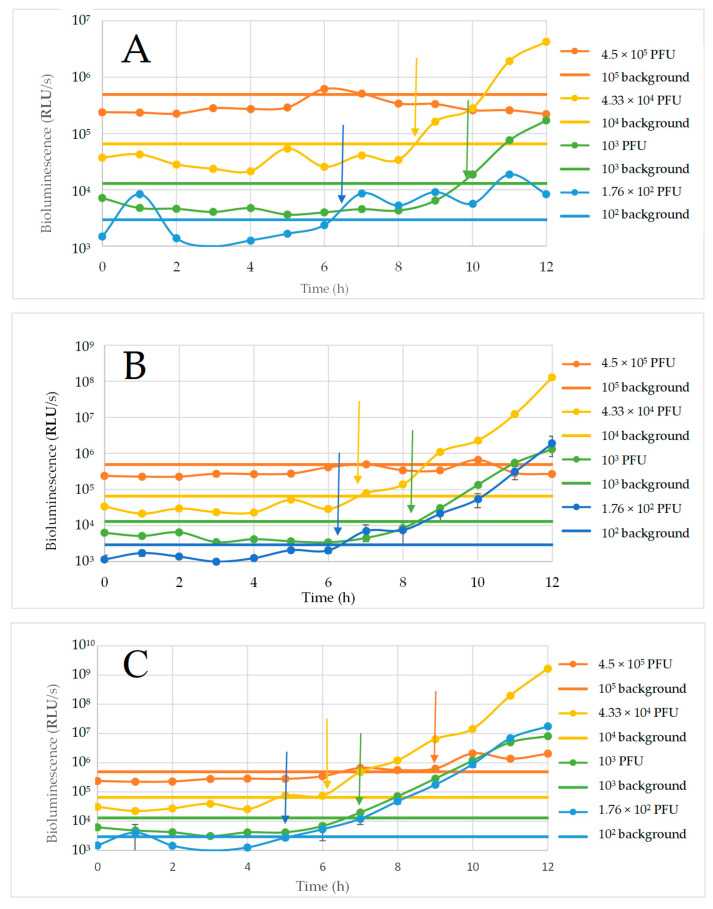
Detection of (**A**) 2 CFU, (**B**) 20 CFU, and (**C**) 200 CFU per sample using various amounts of the ΦV10*nluc* phage. The arrows indicate the times when the luminescence of the sample increases above that of the cell-free controls.

**Figure 2 foods-11-03653-f002:**
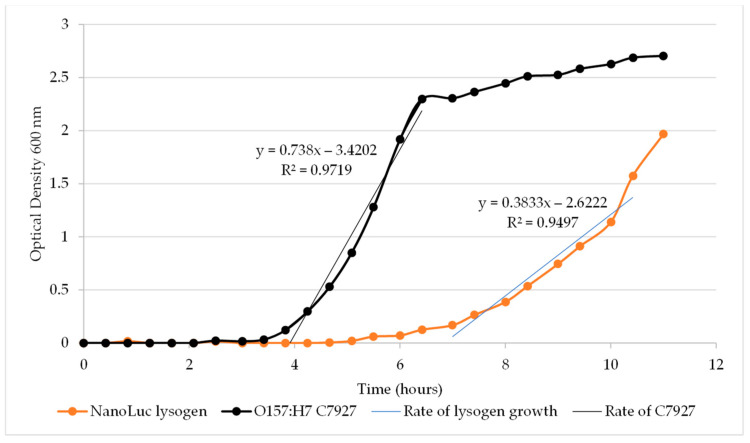
Growth of *E. coli* O157:H7 C7927 and C7927 ΦV10*nluc* lysogen at 37 °C.

**Figure 3 foods-11-03653-f003:**
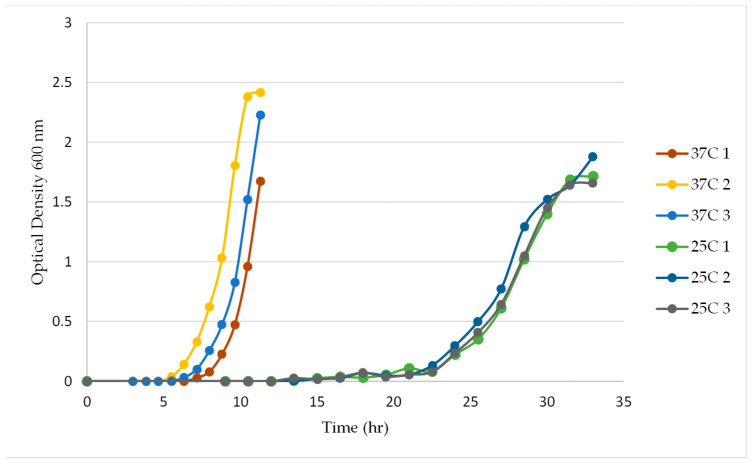
Growth of *E. coli* O157:H7 C7927 ΦV10*nluc* lysogen at 37 °C and 25 °C.

**Figure 4 foods-11-03653-f004:**
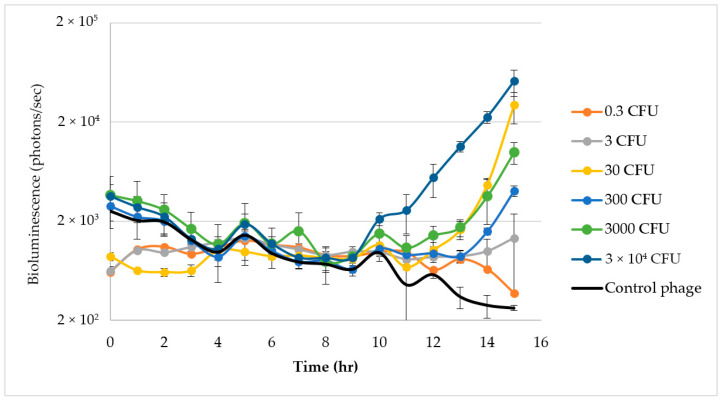
Time course of Luminescence detection of *E. coli* O157:H7 using the standard 1:3 ground beef to mTSB broth ratio in a stomacher bag enrichment format over time.

**Figure 5 foods-11-03653-f005:**
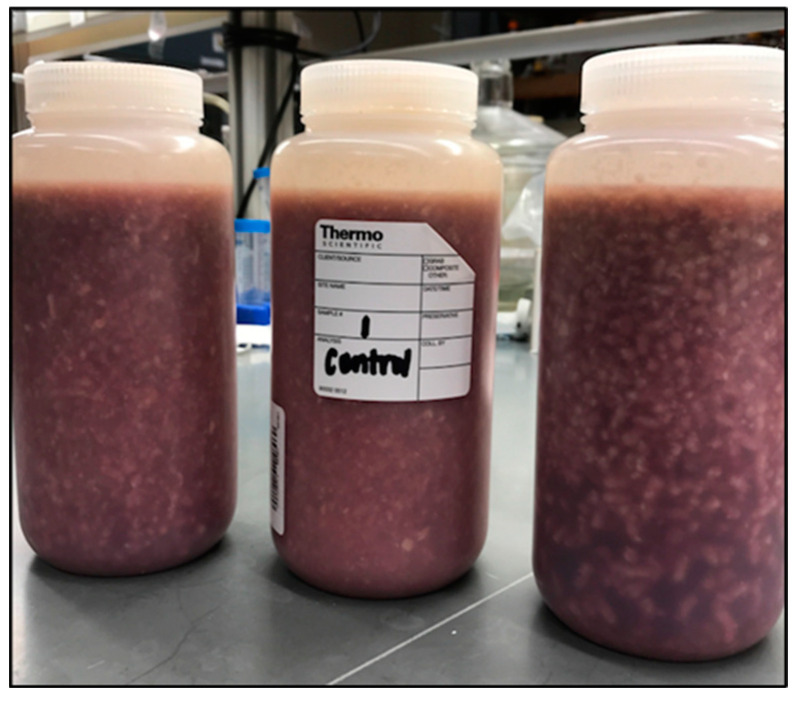
Ground beef enrichment and phage enrichment during shipping. In the examples shown, 650 mL mTSB and 325 g ground beef were added to Nalgene bottles with 1.73 × 10^3^ pfu/mL ΦV10*nluc* (final concentration) for simultaneous enrichment and detection.

**Figure 6 foods-11-03653-f006:**
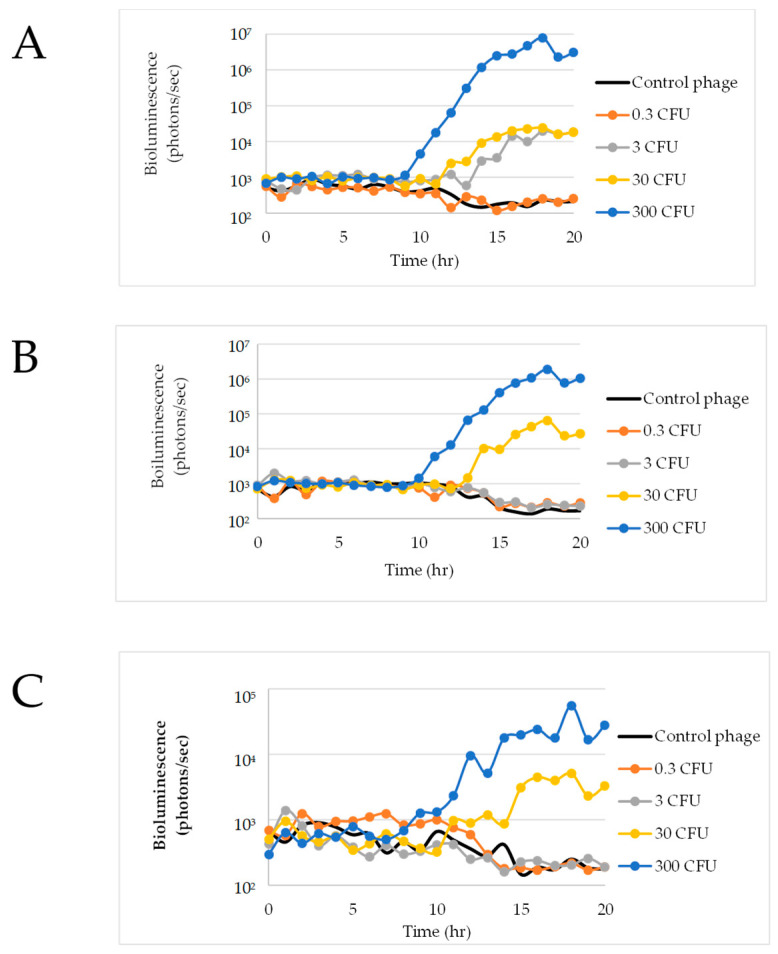
ΦV10*nluc* infectivity in 650 mL mTSB Nalgene bottles. Replicates (**A**) 1, (**B**) 2, and (**C**) 3.

**Figure 7 foods-11-03653-f007:**
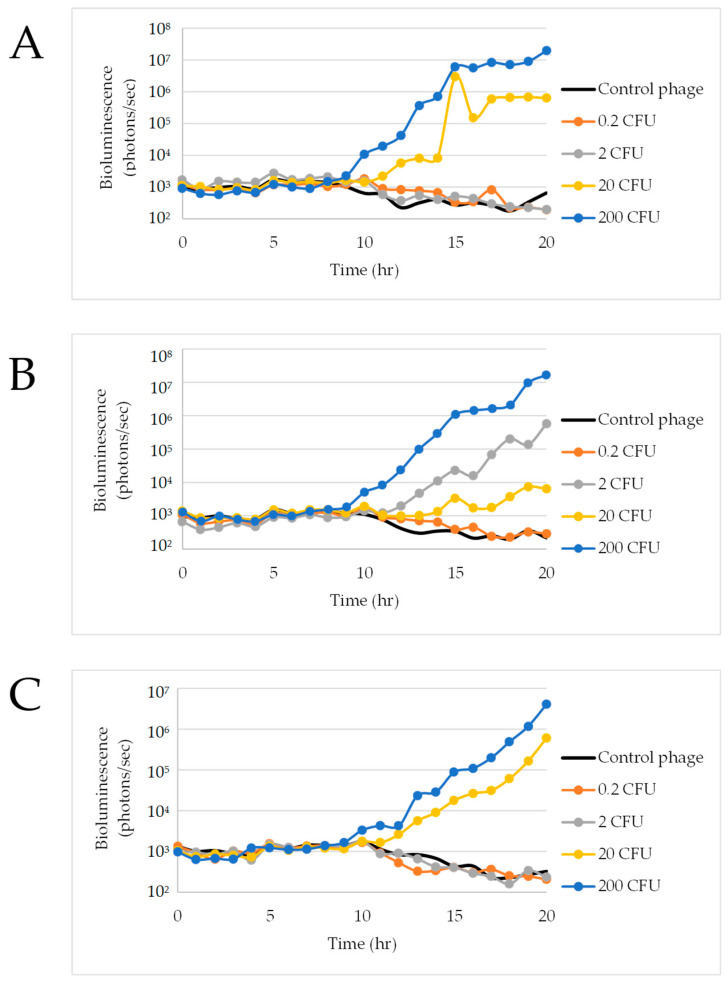
ΦV10*nluc* infectivity in 325 mL mTSB Nalgene bottles. Replicates (**A**) 1, (**B**) 2, and (**C**) 3.

**Table 1 foods-11-03653-t001:** Removal of NanoLuc^®^ Luciferase from ΦV10*nluc* through seven rounds of centrifugal filtration. Luminescence (relative light units per second, RLU/s) was measured for 10 μL of ΦV10*nluc* filtrate after each consecutive centrifugation/filtration cycle for three separate phage preparations.

Centrifugation Cycle	Replicates		
	Filtrate Luminescence (RLU/s)		
	1	2	3
1	419,040,100	371,692,000	349,226,900
2	76,547,500	64,513,800	58,367,400
3	9,490,580	5,744,485	8,108,047
4	382,266	228,219	532,749
5	21,660	16,011	30,201
6	2954	3181	3045
7	3486	4083	2552

**Table 2 foods-11-03653-t002:** Luminescence in ΦV10*nluc* phage preparations before and after purification by centrifugal filtration. Luminescence (RLU/s) was measured in 10 μL of ΦV10*nluc* phage preparation before and after purification using the Amicon^®^ centrifuge filter to remove the NanoLuc^®^ luciferase.

Phage Concentration (pfu/mL)	Luminescence Prior to Purification (RLU/s)	Luminescence After Purification (RLU/s)
7.20 × 10^5^	5.39 × 10^8^	6.08 × 10^3^
1.62 × 10^5^	7.08 × 10^8^	8.08 × 10^3^
1.62 × 10^5^	8.62 × 10^8^	1.38 × 10^4^

## Data Availability

The original contributions presented in the study are included herein.

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
