# Peer review of "Studies on Simultaneous Enrichment and Detection of Escherichia coli O157:H7 during Sample Shipment"

_foods, 2022, doi:10.3390/foods11223653_

Round 1

Reviewer 1 Report

The manuscript shows a study for detection of E coli O157:H7, using a modified phage. They were able to detect a 2 CFU in 2 incubation hours.

The authors shows a good technique to detect E.coli O157:H7 and a I have a couple of comments and suggestions.

In the line 177 there is a mistake, is something like this “-‘to-”, please check it.

Line 111, media preparation, the LB medium was named “Lysogeny broth” and it is Luria-Bertani (LB).

In the figure 1. Please give us a better quality images of the graphs. Please, increase the font’s size in all figures.

The figures 2 and 3 have the caption as a title, please put the caption as a footnote in each figure.

In the figure 3. Please check the “Axis title” I think that it is a mistake.

In the figures 6 and 7, I think that you have to calculate an average for all data. It is very uncommon that someone shows their results as Replicate 1, replicate 2 and replicate 3. You should calculate the average of your replicates and show 1 graph with these results including standard deviation.

Author Response

Foods 1972510 - Responses  Reviewer 1

The manuscript shows a study for detection of E coli O157:H7, using a modified phage. They were able to detect a 2 CFU in 2 incubation hours.

The authors shows a good technique to detect E.coli O157:H7 and a I have a couple of comments and suggestions.

  • In the line 177 there is a mistake, is something like this “-‘to-”, please check it.

RESPONSE: We’re not sure what the reviewer is referring to. Maybe line 22, which is the only occurrence of “to-“ in the manuscript; within the phrase “media-to-sample ratios”.

  • Line 111, media preparation, the LB medium was named “Lysogeny broth” and it is Luria-Bertani (LB).

RESPONSE: We respectfully disagree with the reviewer’s comment.  Lysogeny Broth is the correct name for LB. Giuseppe Bertani himself pointed this out in a Postscript to his 2004 Guest Commentary in the Journal of Bacteriology (Bertani G. 2004. J. Bact. 186:(3):595-600. https://doi.org/10.1128/JB.186.3.595-600.2004 “The acronym has been variously interpreted, perhaps flatteringly, but incorrectly, as Luria broth, Lennox broth, or Luria-Bertani medium. For the historical record, the abbreviation LB was intended to stand for “lysogeny broth.”

  • In the figure 1. Please give us a better quality images of the graphs. Please, increase the font’s size in all figures.

RESPONSE: All the figures were uploaded as high-resolution PDFs as well as the JPGs that were inserted into the text.

  • The figures 2 and 3 have the caption as a title, please put the caption as a footnote in each figure.

RESPONSE: The Figures were shifted to make the correction

  • In the figure 3. Please check the “Axis title” I think that it is a mistake.

RESPONSE:  We thank the reviewers for pointing out the error and this has been corrected in Figure 2.

  • In the figures 6 and 7, I think that you have to calculate an average for all data. It is very uncommon that someone shows their results as Replicate 1, replicate 2 and replicate 3. You should calculate the average of your replicates and show 1 graph with these results including standard deviation.

RESPONSE: The authors appreciate the reviewers comment, but prefer to leave Figures 6 and 7 as presented. The replicates were presented separately to account for fractional positive results expected at very low levels of inoculation. This is exemplified by the 2 CFU and 3 CFU inocula in Figures 6A and 7B, respectively. The single positive result in these experiments may not be a limitation of the detection method but may be due to the likelihood that the experimental contamination of these samples failed to deliver a cell/CFU into the sample. Presenting the data in this way gives a graphical representation of the number of positive results in the three experimental replicates, rather than single line with very large error bars for the fractionally positive results.

Reviewer 2 Report

Reviewer's Comment:

The manuscript describes a rapid system to detect Escherichia coli O157:H7 in meat samples. The paper is well written; however, some minor problems are present.

Introduction

Lines 34-49: please verify the formatting style used, is “aligned with the center” Line 63: “…USDA-FSIS…” the meaning of this acronym is reported later (line 72), please insert here the in extenso name. Lines 62-84: explain better the “government regulatory” limit of detection on the quantity of samples. The paper is too technical. The introduction should explain better the problem in relation to the local government regulations.

Materials and Methods

Lines 111-112: insert the data of the media used (tryptone powder, 1sodium chloride, and yeast), here and in all the paper, where missed (for example kanamycin Line 114; chloroform Line 127; polypropylene glycol Line 131) (Producer, town, state)

Line 119: “E. coli O157:H7 C7927” the strain used is a reference strain. Please explain better. I don’t found information on this strain.

Line 125: “ΦV10nluc lysogen” please insert information on the producer, here and for all the instrument, media and reagent used. (Producer, town, state)

Line 136 and 137: “0.45 μm pore size membrane” and “filter with pore size of 0.1μm” please insert information on the producer, here and for all the instrument, media and reagent used. (Producer, town, state)

Line 125: “Nano-Glo reagent” please insert information on the producer, here and for all the instrument, media and reagent used. (Producer, town, state). The information is reported in the line 166. Please correct.

Line 200: please explain the quantity chosen. 325 g is present also in the introduction, maybe is reported in the

Figure 1: the resolution is low. Please improve if possible

Author Response

Foods 1972510 - Responses Reviewer 2

The manuscript describes a rapid system to detect Escherichia coli O157:H7 in meat samples. The paper is well written; however, some minor problems are present.

Introduction

Lines 34-49: please verify the formatting style used, is “aligned with the center” Line 63: “…USDA-FSIS…” the meaning of this acronym is reported later (line 72), please insert here the in extenso name. Lines 62-84: explain better the “government regulatory” limit of detection on the quantity of samples. The paper is too technical. The introduction should explain better the problem in relation to the local government regulations.

RESPONSE: The formatting has been corrected. The acronyms for USDA and FSIS are now spelled out in their first occurrence in the text on lines 51 and 52. The meaning of the “zero tolerance” for regulated STEC is now given on lines 52-54. This should be sufficient to clarify the government regulatory limit as requested by the reviewer.

Materials and Methods

Lines 111-112: insert the data of the media used (tryptone powder, 1sodium chloride, and yeast), here and in all the paper, where missed (for example kanamycin Line 114; chloroform Line 127; polypropylene glycol Line 131) (Producer, town, state)

RESPONSE: The additions were made in the text as suggested.  We thank the reviewers for pointing this out.

Line 119: “E. coli O157:H7 C7927” the strain used is a reference strain. Please explain better. I don’t found information on this strain.

REPSONSE: Lines 119-120 now have a brief description and a citation to strain C7927. A reference [27] for the strain has been added to the reference list.

Line 125: “ΦV10nluc lysogen” please insert information on the producer, here and for all the instrument, media and reagent used. (Producer, town, state)

RESPONSE: The generation of the ΦV10nluc lysogen of C7927 is described by Zhang et al [reference 22] and is now specially referenced in the text.

Line 136 and 137: “0.45 μm pore size membrane” and “filter with pore size of 0.1μm” please insert information on the producer, here and for all the instrument, media and reagent used. (Producer, town, state)

RESPONSE: The additions were made in the text as suggested.  We thank the reviewers for pointing this out.

Line 125: “Nano-Glo reagent” please insert information on the producer, here and for all the instrument, media and reagent used. (Producer, town, state). The information is reported in the line 166. Please correct.

RESPONSE: The additions were made in the text as suggested.  We thank the reviewers for pointing this out.

Section 2.5 describes the preparation of the Nano-Glo reagent. We have edited the manuscript to direct the reader to this section when we discuss the reagent in this section (2.3.).

RESPONSE: The manufacturer information for the Nano-Glo reagent was inserted after the first occurrence as suggested by the reviewer.

Line 200: please explain the quantity chosen. 325 g is present also in the introduction, maybe is reported in the

The 325 g samples is the size of the regulatory sample tested by USDA-FSIS.  This is now clearly indicated on lines 65-66 in the manuscript.

Figure 1: the resolution is low. Please improve if possible

RESPONSE: All the figures were uploaded as high resolution PDFs as JPGs inserted into the text.